# Power Drill Craniostomy for Bedside Intracranial Access in Traumatic Brain Injury Patients

**DOI:** 10.3390/diagnostics13142434

**Published:** 2023-07-21

**Authors:** Hansen Deng, David J. Puccio, Sharath K. Anand, John K. Yue, Joseph S. Hudson, Andrew D. Legarreta, Zhishuo Wei, David O. Okonkwo, Ava M. Puccio, Enyinna L. Nwachuku

**Affiliations:** 1Department of Neurological Surgery, University of Pittsburgh Medical Center, 200 Lothrop Street, Suite B-400, Pittsburgh, PA 15213, USA; pucciodj3@upmc.edu (D.J.P.); puccioam@upmc.edu (A.M.P.); 2Department of Neurological Surgery, University of California San Francisco, San Francisco, CA 94110, USA; 3Department of Neurological Surgery, Cleveland Clinic Foundation, Cleveland, OH 44195, USA

**Keywords:** craniofacial trauma, craniostomy, external ventricular drain, multimodality monitoring, power drill, traumatic brain injury

## Abstract

Invasive neuromonitoring is a bedrock procedure in neurosurgery and neurocritical care. Intracranial hypertension is a recognized emergency that can potentially lead to herniation, ischemia, and neurological decline. Over 50,000 external ventricular drains (EVDs) are performed in the United States annually for traumatic brain injuries (TBI), tumors, cerebrovascular hemorrhaging, and other causes. The technical challenge of a bedside ventriculostomy and/or parenchymal monitor placement may be increased by complex craniofacial trauma or brain swelling, which will decrease the tolerance of brain parenchyma to applied procedural force during a craniostomy. Herein, we report on the implementation and safety of a disposable power drill for bedside neurosurgical practices compared with the manual twist drill that is the current gold standard. Mechanical testing of the drill’s stop extension (n = 8) was conducted through a calibrated tensile tester, simulating an axial plunging of 22.68 kilogram (kg) or 50 pounds of force (lbf) and measuring the strength-responsive displacement. The mean displacement following compression was 0.18 ± 0.11 mm (range of 0.03 mm to 0.34 mm). An overall cost analysis was calculated based on the annual institutional pricing, with an estimated $64.90 per unit increase in the cost of the disposable electric drill. Power drill craniostomies were utilized in a total of 34 adult patients, with a median Glasgow Coma Scale (GCS) score of six. Twenty-seven patients were male, with a mean age of 50.7 years old. The two most common injury mechanisms were falls and motor vehicle/motorcycle accidents. EVDs were placed in all subjects, and additional quad-lumen neuromonitoring was applied to 23 patients, with no incidents of plunging events or malfunctions. One patient developed an intracranial infection and another had intraparenchymal tract hemorrhaging. Two illustrative TBI cases with concomitant craniofacial trauma were provided. The disposable power drill was successfully implemented as an option for bedside ventriculostomies and had an acceptable safety profile.

## 1. Introduction

Intracranial neuromonitoring and cerebrospinal fluid (CSF) diversion techniques trace back to the earliest days of neurosurgery. External ventricular drain (EVD) placements for neurological emergencies were first described in 1744 by Claude-Nicolar Le Cat, who performed a ventricular puncture followed by a wick placement to enable CSF drainage [1,2]. As defined by the Monro-Kellie doctrine [3], a CSF diversion to decrease intracranial pressure (ICP) is utilized to improve cerebral perfusion and prevent further neurologic deterioration. With advances in cannulation, procedural sterility, and the validation of safe anatomic insertion points, a ventriculostomy today is a lifesaving procedure at the bedside, with diagnostic and therapeutic implications [4].

Common indications include head trauma, subarachnoid hemorrhage, obstructive tumor, and other causes of hydrocephalus. In the traumatic brain injury (TBI) patient population, the current Brain Trauma Foundation guidelines recommend a CSF diversion to treat intracranial hypertension after severe TBI using an antimicrobial-impregnated catheter to prevent infection [5,6]. Incidents of new intracranial hemorrhaging and intracranial infections associated with EVDs range from 5–42% [7,8] and 0–22% [9,10], respectively. Catheter tract hemorrhaging can occur due to multiple cannulations, and extra-axial hemorrhaging or contusions can occur with drill plunging. While a majority of the complications are nonoperative complications and monitored with serial imaging, additional interventions can become necessary after ventriculostomies. Multimodality monitoring can include a combination of ICP, brain tissue partial pressure of oxygen (PbtO_2_), temperature, and microdialysis probes to detect changes in the parenchymal milieu and to guide therapy. For this reason, a ventriculostomy has become a mainstay of neurocritical care for severe TBI, and safe craniostomy techniques can contribute to long-term outcomes [11]. 

The current gold standard for a bedside ventriculostomy involves using the standard manual twist drill included in a cranial access kit. The location of the burr hole is critical for the accurate catheter cannulation of the ventricular space. The length of the drill bit frequently requires calibration using a manual wrench to prevent plunging of the drill bit into the brain parenchyma. Once initiated, if the rotational vector of the drill bit and the entry point are not orthogonal, the downward force from operator may result in tangential skive of the drill bit away from the marked calvarial target, which may cause and be related to concurrent head movement of the patient due to this force, causing mistargeting as well as skull and soft tissue injuries. In the operating room, the aforementioned factors are eliminated with cranial stabilization and the use of high-speed electric surgical drills with readily available power supplies. 

Through a prospective, observational study, the authors have reported on the design, implementation, and safety profile of a disposable power craniostomy drill that can be potentially deployed at the bedside in a United States Level 1 trauma center. Utilizing the power drill technology for cranial access could improve the safety of one of the most frequently performed bedside neurosurgical procedures for diagnostic and therapeutic purposes, and it can also mitigate the overall complication profile. Particularly in patients with difficult anatomy, the power drill could advance patient care through enabling potentially life-saving procedures. A cost analysis was performed, comparing the power drill to the standard manual twist drill. 

## 2. Materials and Methods

### 2.1. Study Design and Ethics

All study patients were enrolled in the University of Pittsburgh Medical Center (UPMC) Brain Trauma Research Center (BTRC) study, which is a prospective observational cohort study conducted at a single Level 1 trauma center of a tertiary care institution (UPMC Presbyterian Hospital, Pittsburgh, Pennsylvania, United States). Patients were consecutively enrolled between July 2021 and July 2022 with the following inclusion criteria: age of 16–80 years, moderate-to-severe TBI defined by an initial Glasgow Coma Scale (GCS) score of ≤13, clinically indicated head computed tomography (CT) scan, and the placement of an EVD and/or a multimodality intracranial monitor placement in a neurotrauma intensive care unit. All study patients provided informed consent prior to enrollment with a legally authorized representative (LAR) due to the severity of the patients’ intracranial injuries. None of the authors here hold any conflicts of interest with the medical device company. The present study was based on the initial experience of implementing this FDA-approved technology in a cohort of patients without altering clinical practices or indications as part of the current standard of care. The surgical drill system is FDA-approved for patient use, and the technology was patented and approved under patent number D784537 (Phasor Health LLC, Houston, TX, USA). All informed consents were obtained as part of the inclusion criteria before the subjects participated in the study. The study protocol was approved by the University of Pittsburgh local Institutional Review Board (IRB #PRO17030027) beginning in June of 2021. The study was conducted in accordance with the Declaration of Helsinki, and the protocol was approved by the Ethics Committee at the University of Pittsburgh.

### 2.2. Demographic and Outcome Measures

Demographic and clinical information (i.e., age, sex, race, GCS score at time of EVD placement, Injury Severity Score (ISS), and mechanism of injury) were collected by research nurse coordinators. The time from presentation to EVD placement, multimodality intracranial monitoring, and duration of EVD management were recorded. All participants received the standard of care neurosurgical treatment at the UPMC, including placements of the EVDs in accordance with the Brain Trauma Foundation guidelines for the management of moderate to severe TBI [12]. Patients requiring craniotomies or decompressive hemicraniectomies for evacuation of intracranial mass lesions and/or the treatment of refractory intracranial hypertension were included. Analyses of the presence of EVD-associated tract hemorrhages on CT were conducted separately by two neurosurgeons. Patients with infectious symptomology for the duration of the EVD or multimodal intracranial monitoring period underwent CSF sampling for the analysis of cell counts and cultures as part of the standard of care for the evaluation of intracranial infection. Two cases with concurrent complex craniofacial trauma are presented in detail. 

### 2.3. Drill Specifications and Operative Technique 

The intracranial access kit used includes a disposable battery powered drill and the ancillary materials needed for an EVD and multimodality monitor placement. The battery pack arrives pre-attached to the handle of the drill. The electric drill is equipped with two buttons (Figure 1). The superior button located on the top of the drill switches between forward and reverse rotations of the drill bit. The inferior button near the handle of the drill is used to operate the drill bit at rotations per minute (RPM) of between 500 and 1000. First, the operator must pull off the plastic tag to allow the battery to engage and power the drill system. The drill stop extension is calibrated in millimeter increments to control for drilling depth (Figure 1). The drill stop has an adjustable depth range of 0–16 mm. It should be twisted to the target depth of 2 mm greater than the outer-to-inner cortical bone thickness as measured on a CT Scan. In patients with outer-to-inner cortical bone thicknesses of >16 mm, the drill stop can be manually detached from the body of the drill using the provided wrench in order to obtain greater drilling depth. 

The drill bit is pre-installed in the power drill. The drill bit’s sizing and the drill’s respective applications to neurological procedures are provided in Table 1. The steps for performing a ventriculostomy are as follows: (1) in a sterile fashion, a 1 cm linear anterior-to-posterior incision is made over the entry point; (2) the operator uses their thumb and index finger of their left hand to center the drill stop extension over the entry point, and the drill bit is placed flush with the outer cortical table of the cranium in order to ensure an orthogonal trajectory; (3) the drill bit is inserted through the drill stop extension to begin engaging the cortical surface; and (4) the operator begins drilling while detecting changes in tactile feedback, which include the bony pitch that is associated with the outer cranial table, cancellous bony, and the inner cranial table. The contents included in an intracranial access kit are illustrated in Figure 2, and they include the following: two Chloraprep applicators, a central service reprocessing wrap, a marker/ruler set, sutures (one 3-0 silk, two 3-0 nylon, and one 4-0 monofilament), a skin stapler, dressing, lidocaine, a blunt-tip needle, and two sterile saline flushes.

### 2.4. Mechanical Testing and Cost Comparison

Eight lot-controlled drill stop extensions from a typical production lot were acquired from Phasor Health, LLC (Houston, Texas, United States) for the analysis of the compressive strength-responsive displacement. Two endpoints were established: (1) the ability to withstand up to 22.68 kilograms (kg) or 50 pounds of force (lbf) for 6 s without displacing more than 0.5 mm; and (2) the incidence of cracks or breaks in the drill stop extensions from the applied force. Samples were evaluated for breakage post-compression using a calibrated tensile tester (LIYI, EVD 10D-50 kg). The extent of displacement was measured using calibrated calipers. For the cost analysis, the authors compared the cost of a standard cranial access kit available at UPMC with the cost of the power drill kit. The cost of items including the chlorhexidine gluconate and isopropyl alcohol skin preps (ChloraPrep™), an extra set of markers and a ruler, the skin stapler, the extra sutures, and the sterile saline flushes that are not included in the standard kit but are included in the new kit were factored into the overall price. 

## 3. Results

Between 4 January 2021 and 4 January 2022, 34 participants with a mean age of 50.7 ± 18.9 years old were included in the present study. The participants’ demographic and clinical characteristics are listed in Table 2. In brief, the most common injury mechanisms were falls and motor vehicle/motorcycle accidents. Trauma registry ISS scores were categorized into minor/moderate (1–15), serious (16–24), severe (25–49), and critical/maximum (50–75). Two patients who experienced pre-hospital cardiac arrest were treated with the appropriate resuscitation and return of spontaneous circulation (ROSC). Admission neurosurgical GCS scores were categorized into moderate (9–12) and severe (3–8).

In the study cohort, the authors elected to use the 5.3 × 70 mm drill bit, as shown in Table 1, which makes a burr hole that can accommodate the diameters of a regular bore EVD (3 mm), a large bore EVD (3.3–3.4 mm), and the the Hemedex quad-lumen bolt system that allows for the placement of a brain tissue ICP probe, an oxygenation probe, a temperature monitoring probe, and a microdialysis catheter to detect various electrolyte levels.

Table 2 provides the EVD and MMM characteristics of the study cohort. The average time from admission to EVD placement was 0.94 ± 1.74 days (range of 0–8 days). A total of 31 (91.18%) patients had an EVD placed within the first two days of admission. Three (8.82%) patients had an EVD placed due to inpatient clinical neurologic worsening 3, 6, and 8 days after admission. Twenty-three patients had a Hemedex quad-lumen bolt placement concurrent with their EVD placement. The mean duration of EVD utilization was 6.90 ± 4.27 days (range of 1–19). In the study cohort, 30 patients underwent surgical decompression that included a craniotomy or a decompressive hemicraniectomy for their TBI. There were three patients who had documented complications from their EVDs and quad-lumen intracranial MMM placements. In the 12 patients who underwent work-up for suspected systemic and/or CNS infection, 1 experienced meningitis with CSF from their EVD growing *Pseudomonas Aeruginosa,* and this patient was appropriately treated with antibiotics (Figure 3). One patient had an intraparenchymal tract hemorrhage related to the EVD placement that required serial CT imaging. One patient had a malfunctioning MMM that required replacement because the Licox brain tissue oxygen monitor no longer provided reliable calibrations. For this reason, it was replaced with a new Licox catheter sterilely at the bedside. The same craniostomy and quad-lumen port were used, without any associated changes in the clinical outcome.

For mechanical testing, each of the eight commercial-grade drill stop extension units were analyzed under 22.68 kg or 50 lbf of compression weight. The axial compressive force was applied for at least 6 s, and the outcome measure was a mechanical displacement greater than or equal to 0.5 mm and signs of crack/breakage, as shown in Figure 4. The average compression extent was 0.18 ± 0.11 mm (range of 0.03 mm to 0.34 mm), as shown in Table 3. The institutional cost of a single unit of the power cranial access kit was obtained at USD 395. The institutional cost of the conventional drill kit was USD 295. The total cost of the conventional cranial access kit in addition to the cost of the ancillary items not included (but that were independently obtained from the medical supplies during EVD/multimodality monitor placement) are provided in Table 4. The total material cost charged at our medical center for the ventriculostomy/MMM placement at the bedside using the conventional unit was USD 330.10 compared to the power kit’s cost of USD 395, with an institutional difference of USD 64.90 per unit. 

### Case Presentations

A 23-year-old male with self-inflicted gunshot wounds (GSWs) to the head with a 0.357 caliber bullet was brought in by EMS with open cranial wounds and orbits head-wrapped. CT imaging showing the multicompartmental intracranial bleed, as well as the severe craniofacial fractures, are depicted in Figure 5A. These injuries included comminuted bilateral frontal calvarial fractures that involved the frontal sinuses and anterior skull base and bone fragments in the bilateral orbits, with a right open-globe injury. The initial GCS score was 7T, and the patient was intubated for respiratory distress. In accordance with the current guidelines on the management of a penetrating TBI, a right-sided EVD was planned at the Kocher’s point [12].

Initially, when using the manual twist drill, the neurosurgeon worsened the depressions of the comminuted fractures with any exertion of downward force to initialize the drilling. The EVD was directly threaded through the splaying of the frontal fractures (Figure 5B), but soon after, the ventricular catheter kinked off at the edge of the cortical table and required replacement. The Phasor power drill was used to create the burr hole 1 cm posterior and lateral from the fracture line (Figure 5C). The EVD was in place for 14 days without malfunctioning. Similarly, a 54-year-old male was found after a self-inflicted GSW with a left frontal entry wound extending through the left occipital lobe (Figure 5D). The initial GCS was 10 T, and with concern for elevated ICP and neurologic worsening from cerebral edema, an EVD was accurately placed (Figure 5E) and assisted with ICP management for 10 days. The patient required a left-sided DHC due to progressive edema and midline shift. 

## 4. Discussion

In a cohort of patients presenting with moderate to severe TBI and the need for invasive neuromonitoring, an electric power drill system was used to safely and efficiently establish intracranial access at the bedside. The progressive innovations over the years in high-speed electric and pneumatic drills have expanded the armamentarium of the neurosurgeon. These tools are largely restricted to the operating room because of portability, disposability, and cost. For such reasons, the manual twist drill continues to be predominantly used in emergent circumstances to access the cranial vault, particularly at the bedside in an emergency room or intensive care unit. Herein, we described the safety and utility of an electric power drill used for bedside craniostomies. We further reported its mechanical advantage, particularly in dealing with complex pathologies. 

A craniostomy performed at the bedside under emergent circumstances is a dynamic procedure. Through a bird’s-eye view, the perpendicular axis of a drill with a cortical surface is maintained during drilling. “Plunging” is a frequent yet under-reported event which can lacerate the dura mater and cause intracranial hemorrhage, which is a complication that a majority of neurosurgeons are reported to have experienced [13]. The overall complication rate of 9% in this study was comparable with rates from previous studies on bedside ventriculostomies. While it is difficult to identify the rate of hemorrhage associated with plunging, Miller et al., in a study that characterized 482 EVDs placed for a spectrum of tumor, bleed, hydrocephalus, stroke, and trauma pathologies, found that the admission platelet count and number of attempts were risk factors for EVD hemorrhaging [9]. In our study, patients on antiplatelet agents were transfused immediately with two units of uncrossed donor platelets based on institutional protocol, as were patients who were identified to have platelet counts of <100,000 µL at the time of admission.

Case 1 highlighted a range of clinical and radiographic factors that the neurosurgeon had to process and to make adjustments. Polytrauma is typically defined as an ISS of 16 or higher, which made up a majority of the patients in the present study. It is known that moderate and severe TBI patients with polytrauma complexes have poorer functional outcomes [14,15,16], and patients with maxillofacial trauma commonly have orbital or nasal fractures along with TBIs [17]. Patients with penetrating TBIs from GSWs are a subpopulation with high rates of complex craniofacial trauma, and this class of patients carries high morbidity and mortality [18,19]. For Case 1, initially with the manual twist drill, the operator created worsening depressions of the comminuted fractures due to any exertion of downward force that would be necessary to initialize drilling. The grip form of the power drill converts drilling into a single-handed procedure [20], freeing the non-dominant hand of the operator to provide further stabilization on the drill stop extension. Additionally, the rotational force from the power drill allows for easier engagement with the cortical skull and reduces the translational force. 

The TBI patient population is susceptible to coagulopathy, although its pathogenesis remains poorly understood [21,22]. Along with poor anatomic landmarks, this further propagates the iatrogenic risk of a blind bedside procedure without hemostatic agents. Similar to improvements in the release mechanism of contemporary perforators to reduce complications [23], the drill stop extension of the power drill is a simple guide that can sustain 50 lb of plunging force without causing depth-plunging of any clinical relevance. As Case 2 showed, TBI patients can have a small ventricular caliber and cerebral edema, which can make cannulating the ventricular system difficult [24]. In our patient, with increased concern for elevated ICP and neurologic worsening from cerebral edema, an EVD was accurately placed to assist with ICP management. The Ghajar Guide was introduced in 1985 to project the ventriculostomy in a perpendicular trajectory to the target of the foramen of Monro in patients with small or slit ventricles [25]. Similar in concept but for the burr hole, the drill stop hubs the skull surface to provide the advantage of preventing the drill tip from skiving and to facilitate perpendicular drilling.

EVD infection rates range from 0 to 22% and infection increases the burden of cost, hospital stay, morbidity and mortality [26,27]. In critically ill patients, the presence of cranial fractures with CSF leaks, craniotomies, and systemic infections have been identified as risk factors for the development of EVD-related infections [28]. In the current study, with its small cohort, the rate of meningitis of 1 in 34 was low compared to rates reported in the literature, and it should be validated with a larger group of subjects as part of a future investigation. While we did not investigate the rates of infection pre -and post-adoption of the antibiotic catheter, our rate of infection was in agreement with the findings from a prior randomized trial showing that positive CSF cultures were seven times less frequent [29]. At our institution, antibiotic catheters are used as the standard of care. While they are more expensive per unit compared to standard catheters, there is robust evidence on their protective benefits and overall healthcare savings [30,31]. Similarly, the unit charges for ventriculostomy kits may vary across medical systems due to a number of factors, and our institutional cost analysis indicated an approximately 19% increased unit cost with the introduction of the disposable power drill.

## 5. Future Directions

While the present study adds to the body of knowledge on bedside craniostomies, it is not without limitations. First, this was a single-center observational study. Factors including race, gender, and sample size parameters limit the generalizability of the data to the overall patient population. Inclusion of a larger cohort from multiple institutions over a longer period of time can further analyze differences in risks for complications and overall outcomes, which will be topics for future investigations. The power drill technology is FDA-approved for use in clinical practice, and extrapolation of the initial findings should take into consideration the small sample population. The estimated cost analysis included in the comprehensive evaluation of the technology is subject to change and can vary by institution. As new technology begins to be adopted into clinical practice, the production costs may vary over time.

## 6. Conclusions

An electric power drill craniostomy system was successfully implemented into bedside neurosurgical practice to assist with ventriculostomies and multimodality intracranial monitoring placements in a prospective cohort of 34 TBI patients presenting to a single United States Level 1 trauma center. The power drill system showed engineering advantages that could potentially enhance the safety profile, reduce the risks of complications, and improve the overall efficiency of common life-saving neurosurgical procedures performed urgently at the bedside, particularly when dealing with difficult anatomy.

## Figures and Tables

**Figure 1 diagnostics-13-02434-f001:**
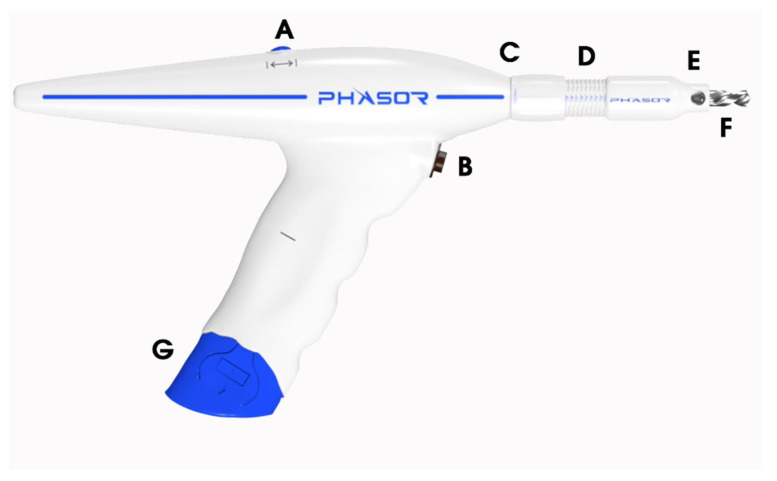
The power drill system. (**A**) The switch toggles between the clockwise versus counter clockwise rotations of the drill bit. (**B**) The switch to begin drilling. (**C**) The distal end of the DSE that is adjusted to the calvalrial thickness. (**D**) The measurements in millimeter increments using the DSE to reach the designated drill depth. (**E**) The proximal end of the DSE that the operator holds flush to the outer cortical bone. (**F**) The drill bit. (**G**) The disposable battery unit. DSE, drill stop extension.

**Figure 2 diagnostics-13-02434-f002:**
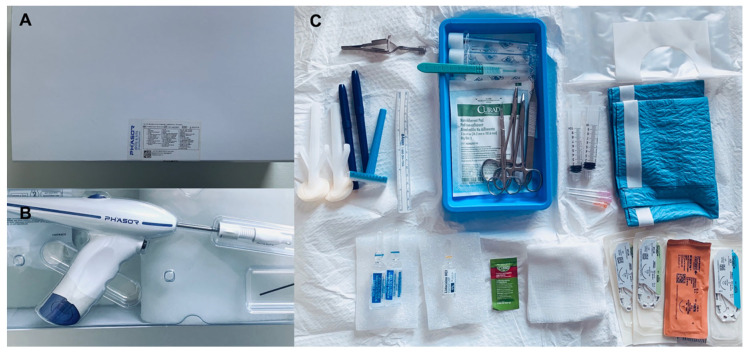
(**A**) Outer packaging. (**B**) Surgical access kit, power drill system, and skin stapler. (**C**) Contents of the surgical access kit once it is opened sterilely in the operative field.

**Figure 3 diagnostics-13-02434-f003:**
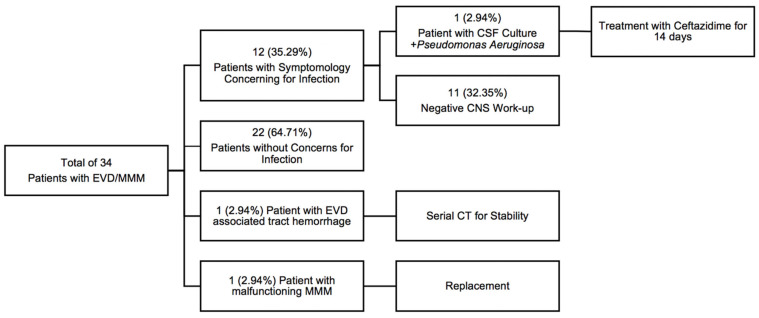
Overall complication profile of the study cohort. EVD, external ventricular drain; MMM, multimodality monitor; CNS, central nervous system; CT, computerized tomography.

**Figure 4 diagnostics-13-02434-f004:**
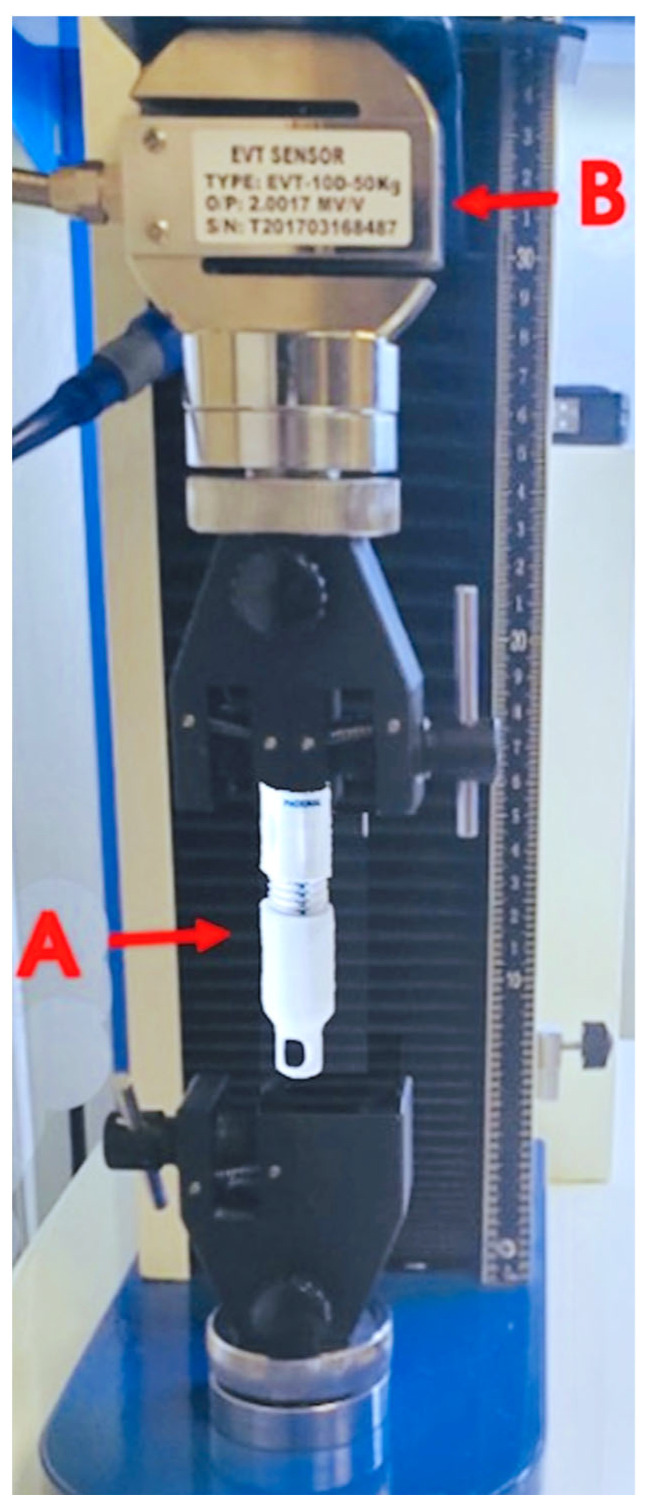
Compression and tensile calibration. Compression and tension force tester with DSE to measure displacement. (**A**) DSE unit. (**B**) The sensor load cell and calibrated calipers for measuring the length pre- and post-exertion of the specified force.

**Figure 5 diagnostics-13-02434-f005:**
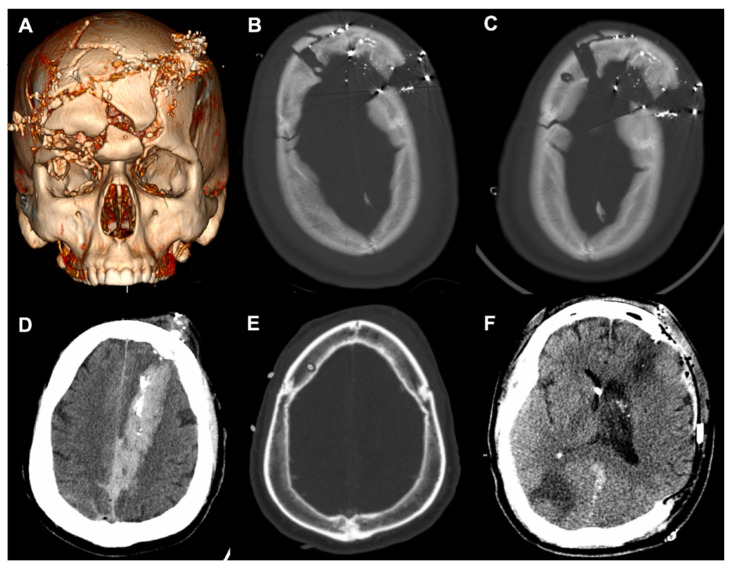
Case presentations of complex craniofacial trauma. (**A**) Three-dimensional reconstruction of Case 1. (**B**) Initial placement of the right frontal EVD through splayed fractures. (**C**) Power drill craniostomy next to the fracture to replace the initial EVD. (**D**) Case 2 demonstrating a penetrating injury with a left frontal entry wound extending into the occipital lobe. (**E**) Right frontal EVD placement using the power drill. (**F**) Axial CT after left-sided decompressive craniectomy. EVD, external ventricular drain; CT, computed tomography.

**Table 1 diagnostics-13-02434-t001:** Drill bit sizing and applications to neurological procedures.

Type of Drill Bit	Diameter (mm)	Maximum Length (mm)	Neurological Procedures
1a	2.4	70	Cervical lateral mass screw cannulation
1b	2.4	210	SEPS placement
2a	2.7	70	Natus Camino and Spiegelberg ICP monitors
2b	2.7	210	Brainlab Varioguide biopsy
3	3.2	210	LITT, SEPS placement, regular EVD catheter (3 mm)
4	4.5	70	Raumedic ICP monitor, regular EVD catheter (3 mm), and large EVD catheter (3.3–3.4 mm)
5 *	5.3	70	Hemedex quad-lumen bolt kit, regular EVD catheter (3 mm), and large EVD catheter (3.3–3.4 mm)
6	5.8	70	SEPS procedure, regular EVD catheter (3 mm), and large EVD catheter (3.3–3.4 mm)

Caption: ICP, intracranial pressure; LITT, laser interstitial thermal therapy; EVD, external ventricular drain; SEPS, subdural evacuating port system; *, the specific size that was utilized in the study cohort.

**Table 2 diagnostics-13-02434-t002:** Demographics and clinical characteristics of the study cohort.

Participant Characteristics	Overall Cohort (N = 34)
Mean age in years (SD, range)	50.7 (18.9, 17–78)
**Sex**	
Male	27 (79.41%)
Female	7 (20.59%)
**Race**	
White	27 (79.41%)
Black	4 (11.77%)
Other	3 (8.82%)
**Mechanism of Injury**	
Fall	15 (44.12%)
Motor vehicle accident	8 (23.53%)
Motorcycle accident	5 (14.71%)
Gunshot wound	4 (11.77%)
Other	2 (5.88%)
**Pre-Hospital Cardiac Arrest**	
Yes	2 (5.88%)
No	32 (94.12%)
**Injury Severity Score**	
Mean (SD, range)	25.50 (6.73, 13–43)
1–15	2 (5.88%)
16–24	9 (26.47%)
25–49	23 (67.65%)
50–75	0
**Initial GCS Score**	
Median (range)	6 (3–10)
3–8	29 (85.29%)
9–12	5 (14.71%)
13–15	0
**Surgical Decompression**	
Yes	30 (88.24%)
No	4 (11.76%)
**EVD**	
Mean time to EVD in days (SD, range)	0.94 (1.74, 0–8)
Mean duration in days (SD, range)	6.90 (4.27, 1–19)
**Quad-lumen MMM**	
Yes	23 (67.65%)
No	11 (32.35%)

Caption: GCS, Glasgow Coma Scale; SD, standard deviation; EVD, external ventricular drain; MMM, multimodality monitor.

**Table 3 diagnostics-13-02434-t003:** Mechanical testing of the compressive displacement using the drill’s stop extension.

Unit #	Before Compression (mm)	During Compression (mm)	Difference (mm)	Signs of Breakage/Crack (Y/N)
1	73.38	73.17	0.21	N
2	73.77	73.71	0.06	N
3	74.18	73.84	0.34	N
4	73.84	73.69	0.15	N
5	73.82	73.79	0.03	N
6	73.81	73.48	0.33	N
7	74.17	73.96	0.21	N
8	74.01	73.89	0.12	N
Overall	73.87	73.69	0.18	N

**Table 4 diagnostics-13-02434-t004:** Estimated cost analysis of the bedside ventriculostomy placement kits.

	Quantity	Unit Cost (USD)	Total Cost (USD)
Material			
Cranial access kit	1	295.00	295.00
Chloraprep applicator	2	3.03	6.06
CSR wrap	1	1.19	1.19
Marker/ruler set	1	2.19	2.19
Suture: 3-0 silk	1	4.41	4.41
Suture: 3-0 nylon	2	2.98	5.96
Suture: 4-0 monofilament	1	6.17	6.17
Skin stapler	1	7.50	7.50
Telfa dressing	1	0.22	0.22
Sterile saline flush	2	0.70	1.40
Total cost using conventional kit			330.10
Total cost using power kit			395.00

Caption: The supply costs were specific to the institution during the study period, and they could vary due to changes over time and with the institution. CSR, central service reprocessing.

## Data Availability

The data presented in this study are available on request from the corresponding author.

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
