# Peer review of "Power Drill Craniostomy for Bedside Intracranial Access in Traumatic Brain Injury Patients"

_diagnostics, 2023, doi:10.3390/diagnostics13142434_

Round 1

Reviewer 1 Report

-The title should be made more concise

-If not included already, performance metrics should be included in the abstract

-Explicit contributions to knowledge should be drilled out as part of Section 1

-The conclusion needs expansions to include reflective comments as well as pathway for further works in the area

n/a

Author Response

Reviewer 1

Comments and Suggestions for Authors

-The title should be made more concise

We thank the Reviewer for their suggestion. We have appropriately shortened the title while maintaining the overall significance as follows:

The original title, “Power Drill Craniostomy for Bedside Intracranial Access: Technique and Application in Traumatic Brain Injury Patients” has been changed to “Power Drill Craniostomy for Bedside Intracranial Access in Traumatic Brain Injury Patients”.

-If not included already, performance metrics should be included in the abstract

We appreciate the Reviewer for their suggestion. The performance metrics are now further highlighted in the abstract:

“Power drill craniostomy was utilized in a total of 34 adult patients with median Glasgow Coma Scale (GCS) score of 6. Twenty-seven (79.4%) patients were male, with mean age was 50.7 years old. The two most common injury mechanisms were falls (44.1%) and motor vehicle/motorcycle accidents (38.2%). EVD was placed in all subjects, and additional quad-lumen neuromonitoring was applied in 23 (67.7%) of patients with zero events of plunging events or malfunctions.”

-Explicit contributions to knowledge should be drilled out as part of Section 1

We thank the Reviewer for their comment. The present study contributes to the existing knowledge of diagnosis and treatment of various common neurological diseases through the modification and application of one of the most neurosurgical tools that are being used at the bedside in hospitals across the country. It is a safer and more efficient tool, particularly in challenging cases that are managed at tertiary centers.

As suggested by the Reviewer, we have added the following to the Introduction to show the explicit contributions to current knowledge from the findings of this investigation:

“Utilizing the power drill technology for cranial access could improve the safety of one of the most frequent neurosurgical procedures performed at the bedside for diagnostic and therapeutic purposes, and also mitigate the overall complication profile. Particularly in patients with difficult anatomy, the power drill could advance patient care through potentially life-saving procedures.”

-The conclusion needs expansions to include reflective comments as well as pathway for further works in the area

We appreciate the Reviewer’s comment. We agree that a discussion on the future direction can be included, as well as expansion of the Conclusion.

We have added the following section, Future Direction, to the manuscript:

“First, given that this was a single-center study, race, gender, and sample size parameters limit generalizability of the data to the overall patient population. Inclusion of a larger cohort from multiple institutions over a longer period of time can further analyze differences in risks of complications, which will be a topic for future investigation.”

We have edited Conclusions as follows:

“The power drill system shows engineering advantages that could potentially enhance the safety profile, reduce the risks of complications, and improve the overall efficiency of common life-saving neurosurgical procedures done urgently at the bedside, particularly when dealing with difficult anatomy.”

Reviewer 2 Report

Introduction:

This critical review evaluates the implementation and safety of a disposable power drill for invasive neuromonitoring procedures in neurosurgery. The study focuses on its application in external ventricular drain (EVD) placement and intracranial monitoring in traumatic brain injury (TBI) patients.

Methodology and Results:

The study conducted mechanical testing, demonstrating the strength and responsiveness of the drill. Cost analysis revealed an estimated increase of $64.90 per unit compared to the manual twist drill. In 34 TBI cases, the power drill was successfully used, with a low incidence of complications (2.9% infection, 2.9% hemorrhage).

Discussion:

While the initial results are promising, this review raises concerns about the study's small sample size and limited evaluation of potential complications. The economic feasibility of adopting the disposable power drill also needs further investigation.

Further research is needed to validate the safety and efficacy of the disposable power drill on a larger scale. Comprehensive analysis should include long-term outcomes and economic considerations before recommending widespread adoption in neurosurgical practice.

Author Response

Reviewer 2

Introduction:

This critical review evaluates the implementation and safety of a disposable power drill for invasive neuromonitoring procedures in neurosurgery. The study focuses on its application in external ventricular drain (EVD) placement and intracranial monitoring in traumatic brain injury (TBI) patients.

Methodology and Results:

The study conducted mechanical testing, demonstrating the strength and responsiveness of the drill. Cost analysis revealed an estimated increase of $64.90 per unit compared to the manual twist drill. In 34 TBI cases, the power drill was successfully used, with a low incidence of complications (2.9% infection, 2.9% hemorrhage).

We appreciate the Reviewer for their time and for their overall support of the findings. Through implementation of this device at single Level I Trauma Center to aid neurocritical care at the bedside, there is the potential for wider adoption of this technology, or other engineering models similar to it, at other hospitals across the country to improve safety and better patient care.

Discussion:

While the initial results are promising, this review raises concerns about the study's small sample size and limited evaluation of potential complications. The economic feasibility of adopting the disposable power drill also needs further investigation.

We thank the Reviewer for their comment. We recognize that one of the limitations was the small sample size of 34 patients with moderate to severe TBI who were treated at a single institution. This was due to the pilot nature of the study and the adoption of a new technology into bedside clinical practice. Secondly, the initial cost analysis showed an estimated $64.90 per unit increase in cost when compared to the standard cranial access kits in use, or 19% increase in cost. This analysis was included as part of an objective evaluation of this new technology, in order to compare its advantages and disadvantages. As seen with other technologies that can potentially improve patient care, the production costs in general decrease significantly over time.

We have added the following to the section, Future Direction:

“The estimated cost analysis included in the comprehensive evaluation of the technology is subject to change and can vary by institution. As new technology begins to get adopted into clinical practice, the production costs may vary over time.” 

Further research is needed to validate the safety and efficacy of the disposable power drill on a larger scale. Comprehensive analysis should include long-term outcomes and economic considerations before recommending widespread adoption in neurosurgical practice.

We appreciate the Reviewer’s comment. The power drill technology is FDA-approved for clinical practice and while this manuscript included a sample size of 34 subjects, it was a pilot study and additional clinical data and economic analysis will be underway and a topic of future investigation.

We have expounded further in the section, Future Direction:

“… a larger cohort from multiple institutions over a longer period of time can further analyze differences in risks of complications and overall outcomes, which will be a topic for future investigation. The power drill technology is FDA-approved for use in clinical practice, and extrapolation of the initial findings should take into consideration the small sample population.”

Reviewer 3 Report

In a cohort of 34 patients with moderate to severe TBI, the authors describe the implementation of a disposable power craniostomy drill for EVD and multimodality intracranial monitoring placement post-trauma, including unwanted side effects.

In principle the data are interesting. For me, the manuscript is partly an advertisement for the device, as a comparable cohort is missing.

Abstract:

Please use metric units, not pounds (also see line 209)

To calculate the % of 1 out of 34 patients with 2.9% is useless.

What about the third patient with side effects? See lines 198-203.

M+M

Ethics Committee at the University of Pittsburgh: please give respective number of permission and date

Is the apparatus purchasable or patented? Please give clear information.

Please always use 3mm or 3 mm (space or no space? throughout the paper and tables).

Line 136: do you really think, that this description is essential for the reader

Results

Table 2: It is not necessary to give identical information twice in text and table. Table alone is ok.

The given % for 27 out of 34 patients with 79.41 is scientifically nonsense (pseudo-accuracy). Please correct all these numbers in your manuscript. In line 275 you give a reasonable number (9%).

Table 1+2+3+4: please delete 1 of the 2 captions,

Table 2: the abbreviation SD is not found in the table.

Table 4: the abbreviation EVD not found in the table; the data of unit costs is not necessary.

Line 251: what is the Phasor power drill – not explained yet.

Discussion:

What is needed is a comparison of the “new” drill with a manual twist drill apparatus in a comparable cohort, and all that in a larger number of patients (see line 315-316). The given results are very preliminary.

Author Response

Reviewer 3

In a cohort of 34 patients with moderate to severe TBI, the authors describe the implementation of a disposable power craniostomy drill for EVD and multimodality intracranial monitoring placement post-trauma, including unwanted side effects.

In principle the data are interesting. For me, the manuscript is partly an advertisement for the device, as a comparable cohort is missing.

We thank the Reviewer for taking their time to review this manuscript and appreciate their interest in the results of the present study. None of the authors here hold any conflict of interest, professional or personal relationship of any nature, with the power drill system Phasor. As is with any innovation in medical technology that can improve upon patient care, especially with recent advances in the field of neuro-endovascular interventional radiology over the past decade, it is not possible to discuss details of new medical technology without disclosing the medical device company. This should not hinder publication of literature related to this topic, or be interpreted as an advertisement for the device. Rather, the objective of the authors was to present their initial experience and preliminary data to readers and colleagues on a new technology that could potentially impact patient care.

We have provided the following statement to the Methods, under section 2.1 Study Design and Ethics:

“None of the authors here hold any conflict of interest with the medical device company. The present study is based on their initial experience of implementing the FDA-approved technology in a cohort of patients without altering clinical practices or indications as part of current standard of care.”

Abstract:

Please use metric units, not pounds (also see line 209)

As suggested by the Reviewer, we included metric units in the manuscript:

Abstract: “…simulating axial plunging of 22.68 kilogram (kg), or 50 pounds of force (lbf) for…”

Methods: “Two endpoints were established: (1) ability to withstand up to 22.68 kilogram (kg), or 50 pounds of force (lbf) for 6 seconds without displacing more than 0.5mm…”

Results: “…analyzed under 22.68 kg, or 50 lbf of compression weight.”

Take 6 seconds out of abstract

We have removed “6 seconds” from the abstract as the Reviewer suggested.

To calculate the % of 1 out of 34 patients with 2.9% is useless.

We recognize that reporting the percentage of complication rates in a pilot study cohort of 34 patients is not as helpful.

We have removed percentages and just reported the incidence of complications in this pilot study as follows:

“One patient developed intracranial infection, and one developed intraparenchymal tract hemorrhage. Two illustrative TBI cases with concomitant craniofacial trauma are provided.”

What about the third patient with side effects? See lines 198-203.

The Reviewer referred to the third patient who experienced a malfunctioning multimodality monitoring that required replacement. In this case, the Licox brain tissue oxygen monitor no longer provided accurate calibrations, and it was replaced with a new Licox catheter at the bedside through the same craniostomy and quad-lumen port without causing any detectable clinical impact.

We have added the following to the Results:

“…had malfunctioning MMM that required replacement. The Licox brain tissue oxygen monitor no longer provided reliable calibrations, and for this reason it was replaced with a new Licox catheter sterilely at the bedside. The same craniostomy and quad-lumen port were used, without any associated change in clinical outcome.”

M+M

Ethics Committee at the University of Pittsburgh: please give respective number of permission and date

We have provided the University of Pittsburgh Institutional Review Board approval and date in the Methods, section 2.1 Study Design and Ethics, as suggested by the Reviewer:

“The study protocol was approved by the University of Pittsburgh local Institutional Review Board (IRB #PRO17030027) beginning in June of 2021.”

Is the apparatus purchasable or patented? Please give clear information.

We thank the Reviewer for their comment. The device technology is FDA-approved as a surgical drill and the patent was approved in April of 2017 (Patent number D784537; Assignee Phasor Health LLC).

We have added this to the Methods section, section 2.1 Study Design and Ethics:

“The surgical drill system is FDA-approved for patient use, and the technology was patented and approved under patent number D784537 (Phasor Health LLC).” 

Please always use 3mm or 3 mm (space or no space? throughout the paper and tables).

We greatly appreciate the Reviewer’s thorough evaluation of the manuscript. As suggested for consistency, we have edited the manuscript to include a space for all of the metrics. These changes are the following:

Lines: 27, 118-120, 155, 183, 212

Table 1

Line 136: do you really think, that this description is essential for the reader

We have reviewed the description that the Reviewer was referring to, and we agree that it can be condensed.

We have shortened the description to the following:

“…4) the operator begins drilling while detecting changes in tactile feedback, which include bony pitch that is associated with the outer cranial table, cancellous bony, and the inner cranial table.”

Results

Table 2: It is not necessary to give identical information twice in text and table. Table alone is ok.

We thank the Reviewer for their comment. We have shortened to manuscript text of the Results section, in order to not be redundant with Table 2:

“Demographic and clinical characteristics are listed in Table 2. In brief, the most common injury mechanisms were falls and motor vehicle/motorcycle accidents. Trauma registry ISS scores were categorized into minor/moderate (1-15), serious (16-24), severe (25-49), and critical/maximum (50-75). Two patients who experienced pre-hospital cardiac arrest who were treated with the appropriate resusci-tation and return of spontaneous circulation (ROSC). Admission neurosurgical GCS scores were categorized into moderate (9-12) and severe (3-8).”

The given % for 27 out of 34 patients with 79.41 is scientifically nonsense (pseudo-accuracy). Please correct all these numbers in your manuscript. In line 275 you give a reasonable number (9%).

We appreciate the Reviewer’s comment. As suggested by the Reviewer previously, we have removed the use of percentage from the text of the manuscript. 

Table 1+2+3+4: please delete 1 of the 2 captions,

We thank the Reviewer for their suggestion. For the captions of Tables 1-4, we have removed the first line of each caption in order to not be redundant with the title of each Table.

Table 2: the abbreviation SD is not found in the table.

For clarification, we utilized the abbreviation SD for the following parameters in Table 2: age, injury severity score, duration of EVD, and hospital stay.

Table 4: the abbreviation EVD not found in the table; the data of unit costs is not necessary.

We thank the Reviewer for their extensive review. We agree with the Reviewer, and we have removed the abbreviation EVD from the caption of Table 4.

The unit cost was important to providing accountability and showing the detailed cost analysis performed at a single institution. These costs may vary by institution, and we have added the following to the caption:

“Supply costs were specific to the institution during the study period, which could vary due to changes over time and by the institution.”

Line 251: what is the Phasor power drill – not explained yet.

The Phasor power drill is a patented and FDA-approved surgical drill that could be utilized for bedside neurosurgical procedures, as compared to traditional the manual twist drill. As suggested in previous comments, we have provided the details in the Methods section of the manuscript:

“The surgical drill system is FDA-approved for patient use, and the technology was patented and approved under patent number D784537 (Phasor Health LLC).” 

Discussion:

What is needed is a comparison of the “new” drill with a manual twist drill apparatus in a comparable cohort, and all that in a larger number of patients (see line 315-316). The given results are very preliminary.

We appreciate the Reviewer for their comment. We recognize that the findings from this study are from a pilot cohort of adult patients, and that these findings need validation with a larger group of subjects as part of future investigation. In this context, the objectives of this investigation are to demonstrate the feasibility and safety of incorporating a new medical technology for one of the most frequent neurosurgical procedures performed at the bedside to assist with the diagnosis and treatment of a spectrum of acute neurological diseases.

As discussed earlier, we have included a Future Directions section in the Manuscript:

“While the present study adds to the body of knowledge on bedside craniostomy, it is not without limitations. First, given that this was a single-center study, race, gender, and sample size parameters limit generalizability of the data to the overall patient population. Inclusion of a larger cohort from multiple institutions over a longer period of time can further analyze differences in risks of complications and overall outcomes, which will be a topic for future investigation. The power drill technology is FDA-approved for use in clinical practice, and extrapolation of the initial findings should take into consideration the small sample population. The estimated cost analysis included in the comprehensive evaluation of the technology is subject to change and can vary by institution. As new technology begins to get adopted into clinical practice, the production costs may vary over time.”

Reviewer 4 Report

Check the names in the authors list. You have 'Nwachuku and MD' at the end. 

The appearance of Figure 2 could benefit from a more polished and professional background. Enhance your photography skills by learning how to capture images with visually distinctive backgrounds, and then utilize various software packages specifically designed for removing unwanted backgrounds. 

Similarly, the blurriness of Figure 4 is quite apparent and it is obviously altered (see those brighter circles, their presence there is almost comical). It is okay to alter the images, but not okay to make it too obvious. It would be beneficial for the authors to enhance their skills in image manipulation techniques.

Shouldn't this be classified as a Technical Report?

Other than that, congratulations on study well done!

Author Response

Reviewer 4

Check the names in the authors list. You have 'Nwachuku and MD' at the end. 

We thank the Reviewer for pointing out this misspelling.

As suggested, we have corrected the last author name to “'Nwachuku MD”.

The appearance of Figure 2 could benefit from a more polished and professional background. Enhance your photography skills by learning how to capture images with visually distinctive backgrounds, and then utilize various software packages specifically designed for removing unwanted backgrounds. 

We appreciate the Reviewer’s comment. We have provided a new Figure 2 that is more polished and with a professional finish for review.

Similarly, the blurriness of Figure 4 is quite apparent and it is obviously altered (see those brighter circles, their presence there is almost comical). It is okay to alter the images, but not okay to make it too obvious. It would be beneficial for the authors to enhance their skills in image manipulation techniques.

We thank the Reviewer for their careful review of the manuscript and the figures. Similar to Figure 2, we have provided an improved Figure 4 to demonstrate the tensile calibration system that was used to evaluate compression force of the drill stop. The bright circles were old labels taken during the testing which are now less obvious in the more polished version of the image.

Shouldn't this be classified as a Technical Report?

We appreciate the Reviewer’s insightful comment. The manuscript entails a comprehensive evaluation of implementing a new medical device that includes a spectrum of clinical, safety and technical outcome measures. Thus, the structure of for a Technical Report would not be a sufficient platform for discussion with these objectives in mind.

Other than that, congratulations on study well done!

We share the Reviewer’s enthusiasm for this investigation. As mentioned before, we think the preliminary findings agree with continuing to utilize the power drill technology for cranial access.  It is one of the most frequent neurosurgical procedures performed at the bedside for diagnostic and therapeutic purposes, and there is potential to mitigate the overall complication profile and improve patient care.

Round 2

Reviewer 3 Report

Dear authors,

thank you for answering me questions and fulfilling my suggestions.

You did a good job.

Author Response

We thank the Reviewer for their warm support of this revised manuscript. Their time and careful review are duly appreciated.